# Circulating Cancer Associated Macrophage-like Cells as a Potential New Prognostic Marker in Pancreatic Ductal Adenocarcinoma

**DOI:** 10.3390/biomedicines10112955

**Published:** 2022-11-17

**Authors:** Christine Nitschke, Benedikt Markmann, Leonie Konczalla, Jolanthe Kropidlowski, Thais Pereira-Veiga, Pasquale Scognamiglio, Martin Schönrock, Marianne Sinn, Marie Tölle, Jakob Izbicki, Klaus Pantel, Faik G. Uzunoglu, Harriet Wikman

**Affiliations:** 1Department of General, Visceral and Thoracic Surgery, University Hospital Hamburg-Eppendorf, 20246 Hamburg, Germany; 2Mildred Scheel Cancer Career Center, University Hospital Hamburg-Eppendorf, 20246 Hamburg, Germany; 3Department of Tumor Biology, University Hospital Hamburg-Eppendorf, 20246 Hamburg, Germany; 4II Medical Clinic and Polyclinic (Oncology), University Hospital Hamburg-Eppendorf, 20246 Hamburg, Germany

**Keywords:** liquid biopsy, Circulating Cancer Associated Macrophage-like cells, pancreatic ductal adenocarcinoma, biomarker

## Abstract

Background: Circulating Cancer Associated Macrophage-like cells (CAMLs) have been described as novel liquid biopsy analytes and unfavorable prognostic markers in some tumor entities, with scarce data for Pancreatic Ductal Adenocarcinomas (PDAC). Methods: Baseline and follow-up blood was drawn from resected curative (*n* = 36) and palliative (*n* = 19) PDAC patients. A microfluidic size-based cell enrichment approach (Parsortix^TM^) was used for CAML detection, followed by immunofluorescence staining using pan-keratin, CD14, and CD45 antibodies to differentiate between CAMLs, circulating tumor cells (CTCs), and leukocytes. Results: CAMLs were detectable at baseline in 36.1% of resected patients and 47.4% of palliative PDAC patients. CAML detection was tumor stage independent. Follow-up data indicated that detection of CAMLs (in 45.5% of curative patients) was an independent prognostic factor for shorter recurrence-free survival (RFS) (HR: 4.3, *p* = 0.023). Furthermore, a combined analysis with CTCs showed the detectability of at least one of these cell populations in 68.2% of resected patients at follow-up. The combined detection of CAMLs and CTCs was also significantly associated with short RFS (HR: 8.7, *p* = 0.003). Conclusions: This pilot study shows that detection of CAMLs in PDAC patients can provide prognostic information, either alone or even more pronounced in combination with CTCs, which indicates the power of liquid biopsy marker analyses.

## 1. Introduction

As a non-invasive approach for cancer diagnosis, prognosis and treatment-monitoring, liquid biopsy has emerged as a technology of great clinical interest in recent years [1,2]. Several liquid biopsy approaches—such as circulating tumor cells (CTCs) analyses—have already shown potential to be valuable biomarkers in many solid cancers [1]. Especially in Pancreatic Ductal Adenocarcinoma (PDAC), liquid biopsy is of great interest as a diagnostic tool in addition to tissue analysis.

Preoperative CTC detection rates in PDAC range from 7% to 42% in curative patients [2,3,4,5]. CTCs have been previously described as independent predictors of poor survival in PDAC, supported by two meta-analyses [3,4,5,6,7].

In most tumors, specialized differentiated macrophages are found within the tissue that can be used as prognostic indicators of tumor invasiveness or suppression. They can acquire expression of stem cell and epithelial markers when they phagocytose apoptotic cancer cells, which is also of interest for liquid biopsy analyses [8,9]. One recently described new tumor-associated cell population in the peripheral blood of tumor patients are Circulating Cancer Associated Macrophage-like cells (CAMLs), also described as tumor-macrophage fusion cell. These cells are often polymorphonuclear, large subtypes (25–300 µm) of circulating cancer-associated cells with myeloid and epithelial phenotypes [10]. Various shapes are described, ranging from small round cells to rod or tailed shapes of large CAMLs [8,11]. An interaction of CAMLs and CTCs within the circulation has been described since CTCs were found to be bound to CAMLs, and there were also CAMLs showing engulfed cells with epithelial phenotypes [10]. Regarding the formation of CAMLs, it needs to be further elucidated whether all atypical cells with macrophage-like as well as epithelial features are from cell fusion, phagocytosis, or direct cell interactions [12].

In various solid tumor entities, CAMLs have already been described as myeloid lineage (CD14+/CD11c+), positive for the blood cell marker CD45, but also cytokeratin positive giant cells with large nuclei found in up to 93% of patients with cancer malignancies [10,13,14,15]. In breast cancer patients, the numbers of CAMLs were higher in metastatic patients—while CAMLs were also observable in 26% of benign lesions, and no CAMLs were found in healthy controls, which suggests their clinical value as a biomarker [13,15]. Furthermore, data from a mice experiment on melanoma showed that the injection of CAMLs was able to generate metastasis [14].

CAMLs, especially when being large, have been described as unfavorable prognostic markers for short recurrence-free survival (RFS) and overall survival (OS) going along with aggressive clinical behavior in different tumor entities [10,16,17]: In metastatic breast cancer patients, high CAML numbers at baseline were significant prognostic markers for poor survival while in non-small cell lung cancer, CAMLs were also independent predictors of short RFS and OS [16,17]. Furthermore, an increase of CAMLs was mainly observed under chemotherapy, suggesting an effect of chemotherapy on CAML release [10].

In PDAC, the role of CAMLs is not sufficiently studied yet. However, there are some published studies about atypical circulating cells with epithelial and macrophage-specific markers that strongly suggest their potentially relevant role in PDAC tumor invasion as well [18].

Our study aims to analyze CAMLs detection and their association with RFS and OS in PDAC patients to evaluate if CAMLs could be a suitable biomarker for prognosis and disease monitoring after tumor resection—alone and in combination with CTCs.

## 2. Materials and Methods

### 2.1. Patient Cohort

In total, *n* = 55 (*n* = 36 curative and *n* = 19 palliative) adult PDAC patients treated at the University Hospital Hamburg-Eppendorf between October 2019 and November 2020 were included in this study. The number of patients considered for inclusion (and checked for the fulfillment of the inclusion criteria) has been described before [19]. From all patients, clinicopathological data were collected. Additional data was provided from the prospectively collected surgical database for patients with pancreatic resections, which was in concordance with the General Data Protection Regulation guidelines. Clinicopathological characteristics (e.g., age, gender, TNM classification), operative details (e.g., duration of surgery, type of resection performed, blood loss), and follow-up data were extracted from the database. The data and patient material collection were approved by the Ethics Committee of Hamburg (PV3548).

### 2.2. Blood Collection

At baseline, before curative surgery was performed or the palliative treatment started, 7.5 mL EDTA blood samples were taken from every patient (*n* = 55). Additionally, peripheral blood samples were drawn from those 22 patients visiting our outpatient department for follow-up (every 3 months post-surgery, *n* = 47) during chemotherapeutic treatment. Blood was taken at least 10 days after last chemotherapeutic treatment. The blood was collected using 7.5 mL EDTA tubes and was processed within two hours after collection via the Parsortix^TM^ Technology (Angle PLC, Surrey, UK)

### 2.3. CAML and CTC Detection

For the combined detection of CAMLs and CTCs in the same blood samples for each patient, we used the marker-independent microfluidic-based Parsortix^TM^ cell separation system. It has been shown in previous studies that this device provides size and deformability-based enrichment by capturing blood cells sized > 6.5 μm [20,21,22]. The data on CTCs has been recently reported [19].

All harvested cells were analyzed via immunofluorescence staining for the nuclear staining DAPI, pan-keratins as an epithelial marker for positive selection, and CD45 as a macrophage-positive marker in CAMLs and for negative enrichment in CTCs. To analyze also CD14 as a myeloid lineage marker in CAMLs, one follow-up sample of each patient and five patient samples at baseline were stained for CD14 alongside DAPI, pan-keratins and CD45. In these additional CD14 stainings, there was no bias in the CAML detection rate observed (atypical cells > 25 µm with one or more nuclei and pan-keratin expression) compared to the whole study cohort. For the immunofluorescence staining, the enriched cell fraction was at first fixed with 4% paraformaldehyde (PFA, Sigma, Ronkonkoma, NY, USA) for 10 min at room temperature. Then, it was permeabilized with 0.2% Triton X-100 (Sigma Aldrich, St. Louis, MO, USA) for 10 min, blocked with 10% AB-Serum (Bio-Rad, Contra Costa County, CA, USA), and incubated with DAPI (1:500), conjugated pan-keratins C11 (1:80, AlexaFluor546 Cell Signalling, USA) and AE1/3 (1:80, Anti-Pan-Cytokeratin AlexaFluor546 Clone (Invitrogen, Waltham, MA, USA), allophycocyanin (APC) conjugated CD45 antibodies (1:150, AlexaFluor647 anti-human CD45 Clone H130 BioLegend, San Diego, MA, USA) and FITC labeled CD14 (1:25, Anti-CD14 Antibody-FITC labeled, MyBioSource, San Diego, MA, USA) for 60 min. The consecutive analysis was performed using immunofluorescence microscopy; atypical cells > 25 µm with one or more nuclei, pan-keratin expression, CD55 expression and—if available—CD14 expression were considered CAMLs.

### 2.4. Statistical Analysis

The statistical analyses were performed using SPSS version 26 (SPSS Inc., Chicago, IL, USA). For the evaluation of a potential association between the CAML status and clinicopathological parameters, as well as CTC status, the χ2 test was used. The impact of CAML status on survival was analyzed by the log-rank test, and the survival curves for patient OS and RFS were plotted using the Kaplan–Meier method. The OS was defined as the time from the date of curative resection or palliative treatment start to either the date of death or last follow-up, whichever occurred first. The RFS was defined as the time from the date of surgical resection in curative patients to the date of recurrence, last follow-up, or date of death, whichever occurred first.

The Cox regression model (Backward Elimination (Wald, Maysville, KY, USA)) was used for multivariate analysis to assess the prognostic value of positivity for mutant-KRAS ctDNA (Table 1). Results are presented as hazard ratio (HR) and 95% confidence interval (CI). Significant statements refer to *p* values of two-tailed tests that were <0.05.

## 3. Results

### 3.1. CAML Detection

At baseline, CAMLs (≥1 CAML/7.5 mL blood) were detected in 40% of all patients (22/55). 36.1% (13/36) of curative (mean 7.3; median 3.0; range 1–27) and 47.4% (9/19) of palliative PDAC patients (mean 13.9; median 6.0; range 1–74) showed CAMLs in their peripheral blood (Figure 1).

After a median follow-up period of 10 months, an overall CAML detection rate of 48% (16/33) was observed (curative: 45.5% (10/22), palliative 54.5% (6/11)) (Appendix A). Due to perioperative death or reduced physical status under therapy, it was not possible to obtain the follow-up blood samples of *n* = 14 curative and *n* = 8 palliative patients. There was no significant difference in the detectability reported at baseline between curative and palliative patients.

Polymorphonuclear CAMLs were observed in 70% of CAML positive patients (67.9% in curative and 70.8% in palliative patients). Very large subtypes of CAMLs (>50 µm) were found in 63.3% of patients with detectable CAMLs (60.7% in curative and 79.2% in palliative patients). There was no significant association with baseline characteristics or survival for both polymorphonuclear and very large subtypes of CAMLs.

In resected curative patients, a significant association of CAML detection during follow-up on OS and RFS was observed (*p* = 0.031 and *p* = 0.003, Figure 2A, B, Appendix A). In the multivariate analysis, CAMLs detection during follow-up was an independent prognostic marker for short RFS (HR: 4.3, *p* = 0.023, Table 2). No association of CAMLs baseline positivity with clinicopathological characteristics (Table 1) or OS and RFS was found. For the group of palliative patients, no significant impact of CAMLs detection at baseline or during follow-up on OS was observed (Appendix A). CAML positivity was independent of ca19-9 increase (Appendix A). Among the relapsed patients, an increase in Ca19-9 before CAML positivity was seen in 14.3% of the patients whereas in 28.6% of patients only a CAML positivity was seen.

### 3.2. Clinical Value of CAMLs alongside CTCs

Our previous CTC findings from the same study cohort showed an overall CTC detection in 25.5% (14/55) of all patients—with 25.0% (9/36—mean 3.0; median 2.0; range 1–6) in curative and 26.3% (5/19—mean 3.6; median 2.0; range 1–11) in palliative patients [19]. After a median follow-up period of 10 months, an overall CTC detection rate of 42.4% (14/33) was observed (curative: 45% (10/22), palliative 36.4% (4/11)) [19].

In the combined analysis of CTCs and CAMLs, there was no association to clinicopathological factors in the detection of CTCs and CAMLs observed at baseline (see Table 1), nor at follow-up (out of 22 patients with follow-up samples, *n* = 7 patients showed nor CALMs or CTCs, *n* = 5 only CAMLs, *n* = 5 only CTCs and another *n* = 5 both). Therefore, CAMLs or CTCs were detectable in 15/22 (68.2%) of curative patients during follow-up.

Nevertheless, the combined analysis of RFS of patients presenting both cells showed an even stronger association with short RFS than the CAMLs analysis alone (*p* = 0.002) (Table 2; Figure 2C). In the multivariate analysis, the joint detection was also an independent prognostic marker for short RFS (HR: 8.7, *p* = 0.003, Table 3). Interestingly, even though the worst outcome was seen for patients with both CAMLs and CTCs, an almost equally bad outcome was seen for patients with CAMLs only. Patients with only classical CTCs seem to do rather good, as their survival resembles those patients in which neither CAMLs nor CTCs were detected (Figure 2D).

## 4. Discussion

Our study shows that the detection of CAMLs in PDAC patients is potentially an unfavorable prognostic biomarker since the detection of CAMLs during follow-up in resected curative patients was associated with significantly shorter RFS and OS. Especially the combined analysis with CTCs might provide additive value for the clinical use of liquid biopsy as a tool for prognosis prediction in PDAC, as at least one of these cell populations was detectable in 68.2% of resected patients at follow-up, and the joint detectability was also significantly associated with short RFS. The HR in the multivariate analysis of the joint analysis of CAMLs and CTCs was higher (HR: 8.7, *p* = 0.003) than in the CAMLs analysis alone (HR: 4.3, *p* = 0.023). This shows the promising potential of combining liquid biopsy approaches for prognosis prediction in PDAC.

Since data on CAMLs detection in PDAC, especially for early-stage patients is still rare, our 40% CAML detection rate at baseline and 48% during follow-up provide valuable data for further liquid biopsy studies in the field of atypical cells [10]. Our detection rates do not reach reported percentages from CAML studies from other solid tumors such as breast cancer and melanoma, where different methods of isolation were applied [10,13,14,15], but our CAML detection rates are similar to reported upper range detection rates from PDAC CTCs studies (about 42% in curative and 48% in palliative patients) which underlines the possible importance of considering CAMLs alongside CTCs in PDAC liquid biopsy research to improve the technology for applicability in the clinics [2,3,4,5]. We were also able to observe different shapes and subtypes of CAMLs, similar to shapes described before (i.e., oblong, amorphous), although there was no significant impact on tumor characteristics or survival of these data [16]. Further studies on these subtypes would be required to describe possible prognostic impacts of shapes and subtypes of CAMLs as atypical circulating cells from patients with solid tumors.

Furthermore, we observed an increase in CAMLs detection during follow-up compared to baseline (45.5% vs. 36.1% in curative and 54.5% vs. 47.4% in palliative patients). As follow-up blood was taken under chemotherapeutic treatment, this observation from our study is in line with the observed increased detectability of CAMLs under chemotherapy reported previously by Adams et al. [10]. The authors have suggested that CAMLs may provide a representation of phagocytosis at the tumor site that could quantify a cell-specific innate immune reaction to the extent of cellular debris caused by chemotherapy [10]. Nevertheless, further studies on the value of these findings are required to understand how increased CAMLs detectability under chemotherapeutic treatment can distinguish between high-risk PDAC patients for early relapse and long-time disease-free survivors.

The survival analysis in the curative patient cohort (with significant results for short OS and RFS) highlights the potential of CAMLs as a significant unfavorable prognostic marker. These results align with previous studies in breast cancer and NSCLC [10,16,17]. In PDAC, reliable biomarkers for prognosis estimation are still rare [23,24]—therefore, liquid biopsy technologies and especially atypical cell analyses—alongside CTCs—are of current research interest for future translation to the clinics [23]. The tumor stage-independent detection of CAMLs is presumably caused by early dissemination during the disease course. CAMLs are assumed to form and disseminate continually during PDAC development and allow subsequent colonization by cell initiating metastasis [18].

The combined analysis of CTCs and CAMLs allows significant conclusions when CTCs in combination with CAML detection are found in follow-up blood. The joint detection of CAMLs and CTCs has an even more substantial prognostic value, regardless of the CTC count [19]. Therefore, the combined approach allows a prognostic prediction in patients showing only low CTC counts. Not only are baseline detection rates of CAMLs higher than those of CTCs (36.1% vs. 25.0% in curative patients), but also the mean cell count of CAMLs at baseline compared to CTCs is 2.4—fold change higher in curative patients. This might also be beneficial for the technological application of this liquid biopsy technology.

Nevertheless, our study has several shortcomings: One major shortcoming is the relatively small sample size and loss of patients for follow-up blood samples, which causes a lack of statistical power for the multivariate model and subgroup analysis by reducing the robustness and reproductivity of results. For survival analysis in palliative patients, the non-significance of the survival results might also be caused by the small patient numbers within the cohort and must be interpreted carefully due to a selection bias. Furthermore, there might also be a selection bias for follow-up samples in curative patients since only patients with a good conditional status might be able to physically present for follow-up blood analysis in the outpatient department.

Since no significant overlapping in the detection of CTCs and CAMLs at baseline nor follow-up was found, both cell analyses might be additively used as a combined approach since at least one of both cell types was found in 68.2% (15/22) during follow-up in curative patients and might be therefore more suitable for a more significant proportion of resected PDAC patients. Especially in combination with the currently used Ca 19-9 and computed tomography during follow-up, liquid biopsy could provide additive value as a prognostic biomarker and individually support postoperative treatment decision-making in PDAC patients.

## 5. Conclusions

The detection of CAMLs in PDAC patients is potentially an unfavorable prognostic biomarker. Especially a combined approach of CAMLs and CTC detection during follow-up under chemotherapy treatment might provide additive value for prognosis prediction in curative PDAC patients independent of Ca19-9 status and might be applicable to the clinics, alongside further approvements of the liquid biopsy technology. Further larger studies are required to investigate the role of circulating CAMLs in PDAC—also in relation to chemotherapy—since they possibly impact the immune response and might serve as a future therapeutic target.

## Figures and Tables

**Figure 1 biomedicines-10-02955-f001:**
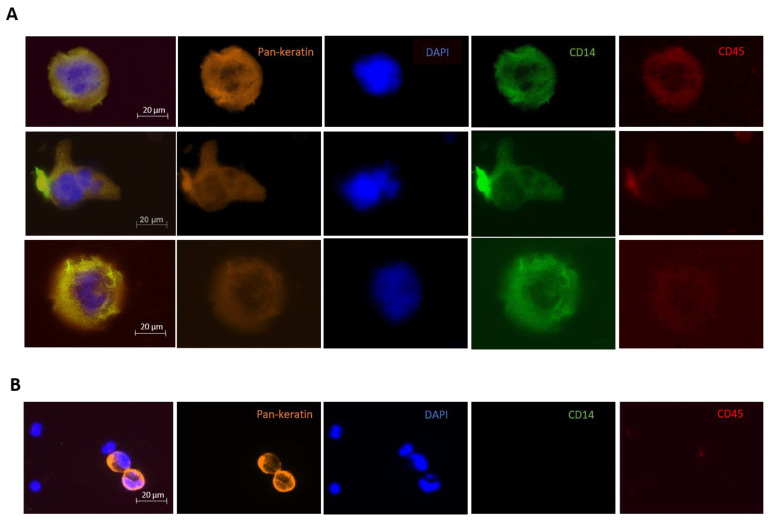
CAMLs and CTCs detected via immunofluorescence staining. (**A**) Different shapes of CAMLs detected in PDAC patients. (**B**) 2 CTCs in a PDAC patient.

**Figure 2 biomedicines-10-02955-f002:**
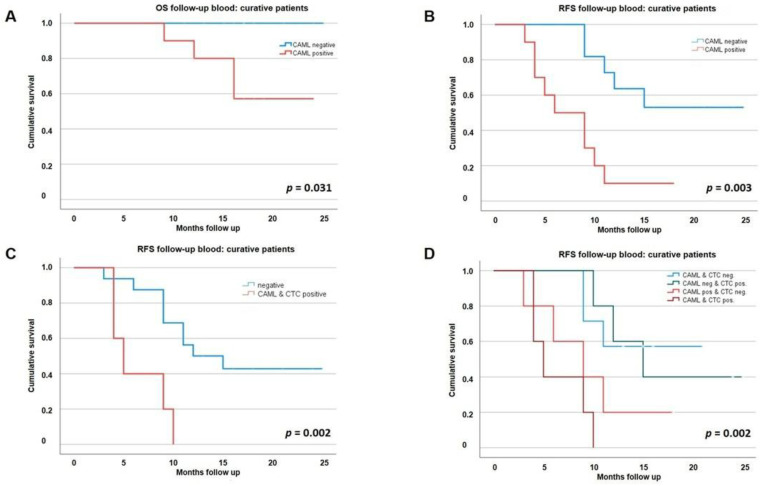
Overall (OS) and recurrence-free survival (RFS) in curative PDAC patients, (**A**) OS influenced by CAML positivity at peripheral follow-up blood in curative patients; (**B**) RFS influenced by CAML positivity at peripheral follow-up in curative patients; (**C**) Combined analysis of CAML and CTC detection: impact of parallel follow-up positivity on RFS in curative patients; (**D**) Combined analysis of CAML and CTC detection: impact of each marker combination.

**Table 1 biomedicines-10-02955-t001:** Correlation of CAML detection at baseline with clinicopathological parameters in curative patients.

		Curative Patients *n* = 36						
		*n*	%	No CAML Detection at Baseline *n* = 23		CAML Detection at Baseline *n* = 13		*p* Value
				*n*	%	*n*	%	
Age	≤67 years	18	50.0	11	47.8	7	53.8	1.000
	>67 years	18	50.0	12	52.2	6	46.2	
Gender	male	17	47.2	11	47.8	6	46.2	1.000
	female	19	52.8	12	52.2	7	53.8	
ECOG	0	20	55.6	11	47.8	9	69.2	0.446
	1	14	38.9	10	43.5	4	30.8	
	2	2	5.6	2	8.7	0	0	
Neoadjuvant treatment	no	29	80.6	19	82.6	10	76.9	0.686
	yes	7	19.4	4	17.4	3	23.1	
Surgical procedure	PD/PPPD	20	55.6	13	56.5	7	53.8	1.000
	left pancreatectomy	13	36.1	8	34.8	5	38.5	
	total pancreatectomy	3	8.3	2	8.7	1	7.7	
Adjuvant treatment	No ^1^	7	19.4	4	17.4	3	23.1	0.686
	yes	29	80.6	19	82.6	10	76.9	
Dindo classification	0–2	19	52.8	13	56.5	6	46.2	0.156
	3–4	12	33.3	7	30.4	5	38.5	
	5	5	13.9	3	13.0	2	15.4	
pT stage	T1-2	17	47.2	11	47.8	6	46.2	1.000
	T3-4	19	52.8	12	52.2	7	53.8	
pN stage	N0	10	27.8	8	34.8	2	15.4	0.270
	N + (N1/2)	26	72.2	15	65.2	11	84.6	
Grading ^2^	G2	23	69.7	14	66.7	9	75.0	0.710
	G3	10	30.3	7	33.3	3	25.0	
R status	R0, CRM-	18	50.0	11	47.8	7	53.8	1.000
	R0, CRM + /R1	18	50.0	12	52.2	6	46.2	
UICC	I-II	28	77.8	18	78.3	10	64.3	1.000
	III	8	22.2	5	21.7	3	62.5	
Ca 19-9	≤500 U/mL	26	72.2	17	73.9	9	69.2	1.000
	>500 U/mL	10	27.8	6	26.1	4	30.8	
	no	16	44.4	9	39.1	7	53.8	
Recurrence	yes	20	55.6	14	60.9	6	46.2	0.493
	no	27	75.0	17	73.9	10	76.9	
CTC detection at baseline	yes	9	25.0	6	26.1	3	23.1	1.000

ctDNA, circulating tumor DNA; ECOG, Eastern Cooperative Oncology Group; CRM, circumferential resection margin; Ca 19-9, Carbohydrate Antigen 19-9; UICC, Union for International Cancer Control; PD, partial pancreatoduodenectomy; PPPD, pylorus preserving pancreatoduodenectomy. ^1^ Not started during follow-up period, or due to reduced ECOG or death. ^2^ For *n* = 3 patients no grading (G) is available.

**Table 2 biomedicines-10-02955-t002:** Univariate analysis on Recurrence-free survival in curative patients.

Univariate		N = 33 ^$^		
Univariate Analyses			Median RFS, Months (95% CI)	*p*-Value
Age	≤67 years	17	10.0 (7.2–12.8)	0.218
	>67 years	16	15.7 (11.0–20.5) *	
Gender	male	16	9.0 (3.8–14.2)	0.170
	female	17	15.0 (9.5–20.5)	
ECOG performance status	0	20	16.0 (9.0–21.8)	0.296
	1	12	11.3 (6.3–14.1)	
	2	1	10.0 (10.0–10.0)	
Uicc stage	I-II	26	16.1 (12.6–19.6) *	0.005
	III	7	6.0 (3.4–8.6)	
R-status	R0; CRM-	17	10.0 (6.0–14.0)	0.227
	R0; CRM + /R1	16	16.0 (9.1–22.9)	
Grading ^~^	G2	21	11.0 (9.6–12.4)	0.353
	G3	9	14.8 (10.3–19.4) *	
Neoadjuvant treatment	no	26	12.0 (6.0–18.0)	0.089
	yes	7	9.0 (0–19.3)	
Adjuvant treatment	no	4	6.0 (0.0–12.0) *	0.009
	yes	29	12.0 (6.7–17.3)	
Clavien-dindo	0–2	20	16.0 (7.9–24.1)	0.161
	3–4	13	10.0 (7.7–12.3)	
Ca 19-9 baseline	<500 U/mL	24	12.0 (5.9–18.1)	0.881
	≥500 U/mL	9	11.0 (8.2–13.8)	
CAMLs detected at FUP ^#^	no	12	18.6 (14.4–22.8) *	0.003
	yes	10	6.0 (1.9–10.1)	
Combined analysis CAML detection and CTC detection at FUP ^#^	no	17	12 (4.7–19.3)	0.002
	yes, both detectable	5	5 (2.9–8.1)	

* median not reached; mean was used; ^#^ FUP number of patients; ^$^ n=3 perioperatively deceased patients not included in the analyses; ^~^ grading not available for *n* = 2 patients; CI, confidence interval; ECOG, Eastern Cooperative Oncology Group; UICC, Union for International Cancer Control; CRM, circumferential resection margin; Ca 19-9, Carbohydrate Antigen 19-9.

**Table 3 biomedicines-10-02955-t003:** Multivariate analysis on recurrence-free survival (RFS) in curative patients.

**Multivariate I: RFS Curative Cohort**		**HR (95% CI)**	***p*-Value**
**Adjuvant treatment**	no	0.4 (0.04–3.5)	0.381
	yes	reference	
**UICC**	no	reference	
	yes	1.9 (1.1–3.4)	0.032
**CAMLs detected at FUP**	no	reference	
	yes	8.7 (2.0–37.0)	0.023
**Multivariate II: RFS curative cohort**		**HR (95% CI)**	***p*-value**
**Adjuvant treatment**	no	0.4 (0.05–3.9)	0.451
	yes	reference	
**UICC**	no	reference	
	yes	2.5 (1.3–4.7)	0.004
**Combined analysis CAML detection and CTC detection at FUP**	no	reference	0.003
	yes	8.7 (2.0–37.0)	

## Data Availability

Data will be available on request from the corresponding author.

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
