# Peer review of "Circulating Cancer Associated Macrophage-like Cells as a Potential New Prognostic Marker in Pancreatic Ductal Adenocarcinoma"

_biomedicines, 2022, doi:10.3390/biomedicines10112955_

Round 1
Reviewer 1 Report
The paper presented by Christine Nitschke et al. investigates the potential use of circulating Cancer Associated Macrophage-Like cells (CAMLs) as prognostic biomarkers in Pancreatic Ductal Adenocarcinomas (PDAC) patients.
The paper is well written and organized and the results could be really informative. Although, as also to the author's statement, the study has a significant limitation in the number of patients tested (only 55), which decreased significantly in the follow-up phase (33). The suggestion would be to collect more data in order to have more significant and relevant results that can also be used actually to correlate CAMLs presence with RFS. The data collected so far are not enough to arrive at the result proposed in the paper.
Another minor concern is the timing of the sampling. It is not clear if CTC and CAMLs were tested on the same blood samples for each patient or at a different time. Moreover, because of the already reported correlation between chemotherapy and CAMLs, it would be appropriate also to verify if the timing between the therapy cycle and the blood draw can influence the detection.
Although this is an interesting study, the sample size is too small to prove the potential use of CAMLs as biomarkers for PDAC patients. If the authors can expand this study to demonstrate the power to detect clinically important differences, resubmission should be suggested.

Author Response
Comments and Suggestions for Authors
The paper presented by Christine Nitschke et al. investigates the potential use of circulating Cancer Associated Macrophage-Like cells (CAMLs) as prognostic biomarkers in Pancreatic Ductal Adenocarcinomas (PDAC) patients.
The paper is well written and organized and the results could be really informative. Although, as also to the author's statement, the study has a significant limitation in the number of patients tested (only 55), which decreased significantly in the follow-up phase (33). The suggestion would be to collect more data in order to have more significant and relevant results that can also be used actually to correlate CAMLs presence with RFS. The data collected so far are not enough to arrive at the result proposed in the paper.
- We thank the reviewer for the response. We agree, that the limitation of the paper is the small sample size, and the loss of patients during the follow-up period. The latter is unfortunately associated with reduced physical conditions of some patients during chemotherapeutic therapy following surgery or chemotherapy given by local oncologist outside the university hospital. The small sample size was clearly stated as a shortcoming. The loss of patients during follow-up has been added as an additional shortcoming in the revised manuscript. Given only 7 days for resubmission does not allow for collection of additional samples.
However, the study submitted is a pilot study, which is indicating a potential impact of CAMLs as negative prognostic marker also in PDAC patients. Despite the small sample size, we were able to show significant results. An expansion of the study needs to be in a multicentric manner to include the number of patients needed in a timely manner.
Another minor concern is the timing of the sampling. It is not clear if CTC and CAMLs were tested on the same blood samples for each patient or at a different time. Moreover, because of the already reported correlation between chemotherapy and CAMLs, it would be appropriate also to verify if the timing between the therapy cycle and the blood draw can influence the detection.
- Thank you very much for this valuable feedback. CTCs and CAMLs were analyzed on the same blood samples for each patient. This was added to the revised manuscript.
Due to the shortcoming of the small sample size, we applied a combined follow-up status of CAML detection at any follow-up time-point. Therefore, the suggested correlation between chemotherapy and CAML (timing between therapy cycle and blood draw) is not possible in this cohort. But we added to the conclusion, that larger studies would also allow for this important analysis.
Although this is an interesting study, the sample size is too small to prove the potential use of CAMLs as biomarkers for PDAC patients. If the authors can expand this study to demonstrate the power to detect clinically important differences, resubmission should be suggested.
- Please see replies above. We have stated the small sample size as a shortcoming of this study, andalso concluded the need for further studies to underline the potential of CAML as a biomarker in PDAC. Within the short time for revision, additional blood collection and follow-up over months / years is not possible.
Reviewer 2 Report
I congratulate the authors for their paper regarding the utility of CTCs and CAMLs as prognostic factors in PDAC.
Overall, this research tries to address the emerging role of "liquid biopsy" techniques in clinical management of pancreatic cancer patients. This is, as I understand a continuation of their CTC-based study in PDAC (Reference#22 - still in press). As a point of consideration, It is hard to assess Novelty/Originality without comparing to the original study (which may include very similar data and analyses).
Regardless, the manuscript has a clear and well delineated design. The analyses are clearly displayed and the conclusions are well supported.
Specific comments:
It is unclear, in the beginning of the manuscript that the aim was to analyze CTCs/CAML during the post-op period - Probably best to better relay that.
What is the role of the palliative group? it seems as though the only data included from it is a supplementary analysis of OS (and the utility of CAML appears contradictory to the curative group).
While Table 1. is very informative for the CAML detection in baseline, a similar table is probably warranted for the post-op CAML.
To better assess the prognostic ability of CAML, can you please include a Kaplan-Meier analysis of other routinely used post-op markers (CEA, CA19-9, Imaging, etc.). This can be included as a supplementary.
In the univariate and multivariate analyses - were the CA19-9 levels also measured at the 3 month time point or are these the pre-op levels?
Can you please include a KM plot of CTCs (to better understand the relative contribution of the combined model).
Author Response
I congratulate the authors for their paper regarding the utility of CTCs and CAMLs as prognostic factors in PDAC.
- We thank the reviewer for the positive response.
Overall, this research tries to address the emerging role of "liquid biopsy" techniques in clinical management of pancreatic cancer patients. This is, as I understand a continuation of their CTC-based study in PDAC (Reference#22 - still in press). As a point of consideration, It is hard to assess Novelty/Originality without comparing to the original study (which may include very similar data and analyses).
- The paper on CTCs has now been published (PMID: 36139565; open access). The paper on CTCs only assessed the clinical value of “classical” CTCs. In this paper we therefore do not assess the clinical value of CTCs alone but only in combination with CAMLs - a new sup-population of cells found in various cancer patients. We therefore feel the two manuscripts do not contain overlapping data. As CTCs are in much studies “classical” liquid biopsy analytes, we did however perform a combined CTC & CAMLs analysis in this paper (see also last point).
Regardless, the manuscript has a clear and well delineated design. The analyses are clearly displayed and the conclusions are well supported.
Specific comments:
It is unclear, in the beginning of the manuscript that the aim was to analyze CTCs/CAML during the post-op period - Probably best to better relay that.
- Thank you for this valuable comment, we have now clarified this issue already in introduction.
What is the role of the palliative group? it seems as though the only data included from it is a supplementary analysis of OS (and the utility of CAML appears contradictory to the curative group).
- Thank you very much for this critical comment. The palliative group was included for comparisons of CAML detection rates at baseline and during follow-up, and also for subtype comparisons. There was no significant difference in detection rates and subtype appearance observed compared to the curative group. This is described in the results section. Indeed, as the reviewer mention, no significant association with OS was found for this group. We believe the very small sample size might have hampered the results. The selection bias has been more highlighted in the discussion of shortcoming in the revised version of the manuscript.
While Table 1. is very informative for the CAML detection in baseline, a similar table is probably warranted for the post-op CAML.
- Thank you very much for this important feedback. We have now added this table as supplementary table 2.
To better assess the prognostic ability of CAML, can you please include a Kaplan-Meier analysis of other routinely used post-op markers (CEA, CA19-9, Imaging, etc.). This can be included as a supplementary.
- Thank you very much for your important feedback. We agree that it would have been of interest to compare the CAML data with other routinely used post-op markers. As imaging is used as the read out for relapse (RFS) the KM plot for imaging can only be done for OS which is clearly of high significance. Regarding CEA, we do not routinely assess CEA on every patient. In our analyses, we used the base line CA19-9 level in the calculations. This was used as for baseline clearly defined cut off for CA19.9 (500U/ml) exist. For post op no such cut off value exist – instead each patients is monitored and an increased is considered as a marker of potential relapse. In the revised manuscript, we have now added to the analyses that baseline CA19-9 values were used.
In the univariate and multivariate analyses - were the CA19-9 levels also measured at the 3 month time point or are these the pre-op levels?
- See above.
Can you please include a KM plot of CTCs (to better understand the relative contribution of the combined model).
- Thank you for this valuable comment. We have now included a new KM plot in which, all combinations of CTCs and CAMLs can be seen. As shown in the new figure 2D, the worst survival is seen for patients with both CAMLs & CTCs, whereas almost equally bad survival is seen for patients with CAMLs only. Patients with only classical CTCs seem to do rather good as their survival resembles those patients, in which neither CAMLs nor CTCs were detected.
Round 2
Reviewer 1 Report
Dear Author,
As for my first round, the study you conducted could be really informative and useful. However if you want to submit as it is, with the low number of samples, it will have to be structure as a pilot study. The preliminary status has to be reflected int the title ad well as in all the affirmations done in the results and conclusion section. I made some comments on the pdf file that are only some of what need to be revised.
Therefore, i advised the editor that the paper need to be revised in his form and resubmitted after major revisions.
Author Response
Reviewer 1
As for my first round, the study you conducted could be really informative and useful. However if you want to submit as it is, with the low number of samples, it will have to be structure as a pilot study. The preliminary status has to be reflected int the title ad well as in all the affirmations done in the results and conclusion section. I made some comments on the pdf file that are only some of what need to be revised.
Therefore, i advised the editor that the paper need to be revised in his form and resubmitted after major revisions.
- We agree that based on the rather small sample size the title could be changed. We have therefore now changed the title to “Circulating Cancer Associated Macrophage-Like cells as potential a new prognostic marker in Pancreatic Ductal Adenocarcinoma”. In addition, we have included the word pilot in the abstract and modified the conclusions slightly. We did already have a complete section with shortcomings “Nevertheless, our study has several shortcomings: One major shortcoming is the relatively small sample size and loss of patients for follow-up blood samples, which causes a lack of statistical power for the multivariate model and subgroup analysis, by reducing robust-ness and reproductivity of results. For survival analysis in palliative patients, the non-significance of the survival results might be also caused by the small patient numbers within the cohort and have to be interpreted carefully due to a selection bias. Furthermore, there might also be a selection bias for follow-up samples in curative patients since only patients with a good conditional status might be able to physically present for follow-up blood analysis in the outpatient department”. We do not know what else should be included.
- Of note, many studies in the field of liquid biopsy are based on rather small number of samples especially when talking about more rare cancer entities. We identified 9 studies on CAMLs, out of which 5 had between 30-70patients. Only one study used the word pilot (PMID: 27706044, PMID: 33148307, PMID: 32798130, PMID: 31771043, PMID: 36230499).
- Comments to the questions raised in the pdf:
- CTC and CAML analyses were done simultaneously as clearly stated in materials and methods
- Blood was taken at least 10 days after last ctx. Info has now been included in the revised manuscript.
- Comments to small sample size – see above.

Reviewer 2 Report
I thank the authors for the opportunity to review their revised manuscript
Most of the previous comments have been nicely addressed. The post-op CAML supplementary table 1 has only N=22, why is that? I would have expected an N=33. Additionally the Adjuvant therapy numbers are puzzling - it seems as though the vast majority did not receive adjuvant therapy - Can you please comment on that.
The authors replied that "In our analyses, we used the base line CA19-9 level in the calculations. This was used as for baseline clearly defined cut off for CA19.9 (500U/ml) exist. For post op no such cut off value exist – instead each patients is monitored and an increased is considered as a marker of potential relapse. In the revised manuscript, we have now added to the analyses that baseline CA19-9 values were used."
However, it is difficult to assess the relative prognostic value of a novel metric such as CTCs/CALMs without comparing to other used markers such as CA19-9 elevation. If such a comparison is unavailable, it should be mentioned in the limitations of interpretation of the data.
Another minor comments: in the Multivariate analysis, what does UICC yes/no mean? Is this supposed to be I-II vs. III ?
Author Response
Reviewer 2
I thank the authors for the opportunity to review their revised manuscript
Most of the previous comments have been nicely addressed. The post-op CAML supplementary table 1 has only N=22, why is that? I would have expected an N=33. Additionally the Adjuvant therapy numbers are puzzling - it seems as though the vast majority did not receive adjuvant therapy - Can you please comment on that.
- Thank you very much for this important feedback. The aim of the new supple-mentary table 1 was to show the clinicopathological correlation with CAML detection during follow-up. We therefore included only those patients with a follow-up CAML status (n=22) for a clinicopathological correlation (not all patients from baseline (n=36) nor all patient minus the perioperative deceased ones (n=36-3=33)). The loss of patients during the follow-up period was stated in the limitations. The latter is unfortunately associated with reduced physical conditions of some patients during chemotherapeutic therapy following surgery.
- We are very sorry, indeed there was a mistake in 2 of the tables where the number of persons receiving or not receiving adjuvant therapy was mixed. We have now corrected the new supplementary table and table 1. Thank you for this good observation.
The authors replied that "In our analyses, we used the base line CA19-9 level in the calculations. This was used as for baseline clearly defined cut off for CA19.9 (500U/ml) exist. For post op no such cut off value exist – instead each patients is monitored and an increased is considered as a marker of potential relapse. In the revised manuscript, we have now added to the analyses that baseline CA19-9 values were used."
However, it is difficult to assess the relative prognostic value of a novel metric such as CTCs/CALMs without comparing to other used markers such as CA19-9 elevation. If such a comparison is unavailable, it should be mentioned in the limitations of interpretation of the data.
- We agree that changes in Ca19-9 levels gives important clinical information. We have now checked the elevation of Ca19-9 in the FUP samples. The results shows that also post Ca19-9 values are independent of CAML data similarly to the pre OP value. We have now added this data as a supplementary table 3 and included in the results on page 6.
Another minor comments: in the Multivariate analysis, what does UICC yes/no mean? Is this supposed to be I-II vs. III ?
-Thank you very much for this good observation as well. You are right, it is supposed to be UICC stages I-II vs. III as in the univariate analyses. We have now corrected it.

Round 3
Reviewer 1 Report
Dear Atuhors,
Thank you for addressing my quetions and comment. The study is now presented as a pilot study on preliminary data. In the current form I advise to acept the paper in the current form.
Author Response
we thank you very much for this comment